# Atypical lymphocytes in the peripheral blood of COVID-19 patients: A prognostic factor for the clinical course of COVID-19

**Jun Sugihara**[1,2]\*, **Sho Shibata**[1,2], **Masafumi Doi**[1], **Takuya Shimmura**[1], **Shinichiro Inoue**[1], **Osamu Matsumoto**[3], **Hiroyuki Suzuki**[3], **Ayaka Makino**[3], **Yasunari Miyazaki**[2]

1 Department of Respiratory Medicine, Kashiwa Municipal Hospital, Kashiwa, Chiba, Japan, 2 Department of Respiratory Medicine, Tokyo Medical and Dental University, Bunkyo-ku, Tokyo, Japan, 3 Clinical Laboratory, Kashiwa Municipal Hospital, Kashiwa, Chiba, Japan

\* sugihara.pulm@tmd.ac.jp

**Data Availability Statement:** All relevant data are within the manuscript and its Supporting Information files.

## Abstract

### Background

Clinical observations have shown that there is a relationship between coronavirus disease 2019 (COVID-19) and atypical lymphocytes in the peripheral blood; however, knowledge about the time course of the changes in atypical lymphocytes and the association with the clinical course of COVID-19 is limited.

### Objective

Our purposes were to investigate the dynamics of atypical lymphocytes in COVID-19 patients and to estimate their clinical significance for diagnosis and monitoring disease course.

### Materials and methods

We retrospectively identified 98 inpatients in a general ward at Kashiwa Municipal Hospital from May 1st, 2020, to October 31st, 2020. We extracted data on patient demographics, symptoms, comorbidities, blood test results, radiographic findings, treatment after admission and clinical course. We compared clinical findings between patients with and without atypical lymphocytes, investigated the behavior of atypical lymphocytes throughout the clinical course of COVID-19, and determined the relationships among the development of pneumonia, the use of supplemental oxygen and the presence of atypical lymphocytes.

### Results

Patients with atypical lymphocytes had a significantly higher prevalence of pneumonia (80.4% vs. 42.6%, p < 0.0001) and the use of supplemental oxygen (25.5% vs. 4.3%, p = 0.0042). The median time to the appearance of atypical lymphocytes after disease onset was eight days, and atypical lymphocytes were observed in 16/98 (16.3%) patients at the first visit. Atypical lymphocytes appeared after the confirmation of lung infiltrates in 31/41

**Funding:** This study is funded by Japan Agency for Medical Research and Development. S. S. received this award. Grant number is 21jk0210034h0002. URL of the funder is https://www.amed.go.jp/en/index.html The funder had no role in study design, data collection and analysis, decision to publish, or preparation of the manuscript.

**Competing interests:** The authors have declared that no competing interests exist.

(75.6%) patients. Of the 13 oxygen-treated patients with atypical lymphocytes, approximately two-thirds had a stable or improved clinical course after the appearance of atypical lymphocytes.

## Conclusion

Atypical lymphocytes frequently appeared in the peripheral blood of COVID-19 patients one week after disease onset. Patients with atypical lymphocytes were more likely to have pneumonia and to need supplemental oxygen; however, two-thirds of them showed clinical improvement after the appearance of atypical lymphocytes.

## Introduction

Coronavirus disease 2019 (COVID-19), an emerging disease caused by a coronavirus, has imposed a substantial health burden worldwide. Since the first report in December 2019, there have been approximately 149 million cases of COVID-19 and approximately 3 million related deaths worldwide as of April 29, 2021 [1]. COVID-19 has a broad spectrum of manifestations, ranging from asymptomatic cases or cases of mild symptoms similar to those of the common cold to cases of severe respiratory disease with systemic inflammation and thrombosis. The clinical characteristics hinder its prompt diagnosis and the prediction of the clinical course of the disease, and many efforts to clarify the characteristics of this disease have been made.

Atypical lymphocytes are large lymphocytes with varied morphology in the peripheral blood of patients with several viral infections, such as Epstein-Barr virus infection, cytomegalovirus infection, rubella, Hantavirus infection, viral hepatitis and HIV infection [2]. Several studies have shown that atypical lymphocytes have also been found in the peripheral blood [3–5] and bronchoalveolar lavage (BAL) samples [6] of COVID-19 patients. It may reflect the disease pathophysiology and provide important information about the diagnosis or prognosis of the disease.

However, few studies have reported the clinical significance of atypical lymphocytes in COVID-19 patients. A retrospective analysis surveying data from patients' first visits demonstrated that a positive result on a severe acute respiratory syndrome coronavirus 2 (SARS-CoV-2) polymerase chain reaction (PCR) was related to a significantly higher prevalence of atypical lymphocytes than a negative PCR result [7], and another study comparing COVID-19 patients hospitalized in the intensive care unit (ICU) and non-ICU wards indicated that atypical lymphocytes were associated with a milder clinical course of the disease [8]. However, there is limited information about the time course of the number or proportion of atypical lymphocytes in the peripheral blood. Such information would be useful for estimating the clinical course of COVID-19.

Therefore, we investigated the relationship between atypical lymphocytes and the clinical course of COVID-19 and aimed to demonstrate the clinical significance of that relationship in the present study.

## Materials and methods

### Study design and patients

This is a retrospective cohort study. We enrolled consecutive COVID-19 patients admitted to Kashiwa Municipal Hospital from May 1st, 2020, to October 31st, 2020, in this study. Their

diagnoses were confirmed with positive results on reverse-transcription PCR assays for SARS-CoV-2. All patient admissions were determined by regional public health centers because they were responsible for determining which COVID-19 patients should be hospitalized and which should recuperate at home in Japan. Each hospital was assigned the task of caring for patients with mild-to-moderate COVID-19 or patients with severe COVID-19 (requiring ICU care) based on the capacity. Our hospital was assigned to care for patients with mild-to-moderate COVID-19. If the condition of the patients worsened and they required ICU care, we transferred them to the appropriate hospitals.

## Data collection

We collected the clinical data pertaining to the enrolled patients from the medical records at Kashiwa Municipal Hospital. The collected data were demographics, symptoms before admission, comorbidities, blood test results, radiographic findings, treatment after admission and the clinical course.

The symptoms included fever (defined as temperature $\geq 37.5°C$), upper respiratory symptoms (sore throat, rhinorrhea and nasal congestion), lower respiratory symptoms (cough, sputum and dyspnea), constitutional symptoms (fatigue, myalgia, arthralgia, headache and chills), olfactory or taste dysfunction, and loose stool. We also inquired the day the symptoms had started, which was designated the day of the onset of disease.

The comorbidities included hypertension, diabetes mellitus, chronic heart disease, chronic obstructive pulmonary disease (COPD), chronic kidney disease, coronary vascular disease, cerebrovascular disease and malignancy.

The blood tests included a complete blood count with white blood cell (WBC) differential, lactase dehydrogenase (LDH), total bilirubin (T-Bil), urea nitrogen (UN), creatinine and C-reactive protein (CRP). These tests were performed as part of routine clinical care. The complete blood count with WBC differential was performed with an XT-2000i hematologic analyzer (Sysmex Co., Kobe) for all samples. When the analyzer detected the existence of atypical lymphocytes, a laboratory technician (O.M., H.S. or A.M.) made a blood smear and examined it microscopically to confirm the morphological abnormalities and differential count. For those patients with abnormal lymphocytes, we also obtained the percentage of atypical WBCs, the time of the appearance of the atypical lymphocytes and whether they were present at the first visit.

The computed tomography scans were evaluated to determine the radiographic findings. Pulmonologists (J.S., T.S., M.D. and S.S., with eight, eight, nine, and twelve years of experience, respectively) reviewed the images and interpreted whether they had radiographic evidence of pneumonia compatible with COVID-19.

The treatments were supplemental oxygen, antivirals, and corticosteroids. The concentration of oxygen was also collected.

## Ethics issues

This study was approved by the ethics committee of Kashiwa Municipal Hospital (#02–5, Feb 18th, 2020). The need to obtain written informed consent was waived because of the anonymous nature of the data. Instead, we announced the study officially and ensured that patients could opt out of the study.

## Statistical methods

Categorical variables are expressed as numbers with percentages, and continuous variables are expressed as medians with interquartile ranges. Statistical analyses were conducted as follows:

the Shapiro-Wilk test was used to assess the normality of the variable distributions; Student's t test was used for comparisons of parametric continuous variables; one-way analysis of variance (ANOVA) was used for comparison of continuous variables among more than three groups; the Wilcoxon rank-sum test was used for comparisons of nonparametric continuous variables; and Fisher's exact test was used to assess the ratios of categorical variables. All tests were conducted with R software. P values $< 0.05$ were considered statistically significant.

## Results

### Patient characteristics

A total of 98 patients were enrolled in the study, and we divided them into two groups based on the presence of atypical lymphocytes in the peripheral blood. The morphologies appeared as previously reported in papers [3–5] (representative cells are shown in Fig 1). Patient characteristics are shown in Table 1. All patients were discharged alive. The distributions of basic demographic features, such as age, sex and comorbidities, were not significantly different between those with and without atypical lymphocytes. However, patients with atypical lymphocytes tended to have a longer duration between symptom onset and their first visit. Regarding the signs and symptoms, while patients without atypical lymphocytes were more likely to be asymptomatic, patients with atypical lymphocytes were significantly more likely to have fever at the first visit. They also had a significantly higher likelihood of radiographic evidence of pneumonia and the need for supplemental oxygen than those without atypical lymphocytes. The distribution of atypical lymphocyte fractions is shown in Fig 2. Forty-seven out of 98 (48%) patients had a blood smear that did not show atypical lymphocytes. There were 9 patients with 1% atypical lymphocytes. Twelve and 13 patients had 2% and 3% atypical lymphocytes respectively, while 8 and 7 patients had 4 and 5% atypical lymphocytes, respectively. A higher fraction (7 and 8%) of atypical lymphocytes was observed in only 2 patients (one for each fraction). The date of the first identification of atypical lymphocytes after disease onset is shown in Fig 3. The median time from symptom onset to the appearance of atypical lymphocytes was eight days. The majority of the patients with atypical lymphocytes were detected 5–9 days after disease onset and peaked at nine days. At the first visit, atypical lymphocytes were found in 16/98 (16.3%) patients, and 5/61 (8.2%) patients with radiographic evidence of pneumonia did not have any pulmonary infiltration.

### Radiographic features

In COVID-19 patients, the observation of radiographic features characteristic of pneumonia tended to precede that of atypical lymphocytes. A total of 41 patients had both radiographic evidence of pneumonia and atypical lymphocytes. Among them, radiographic evidence of pneumonia was observed first in 31/41 (75.6%) patients, atypical lymphocytes were observed first in 2/41 (4.9%) patients, and both were observed on the same day in 8/41 (19.5%) patients. Fig 4A shows the time intervals between the appearance of the radiographic evidence of pneumonia and that of atypical lymphocytes for each patient.

The distributions of time of the first identification of atypical lymphocytes after disease onset are shown in Fig 4B. The median time from disease onset to the detection of atypical lymphocytes was nine days and that to the observation of radiographic evidence of pneumonia was five days, and the distributions were significantly different (p $< 0.0001$ by Wilcoxon rank-sum test).

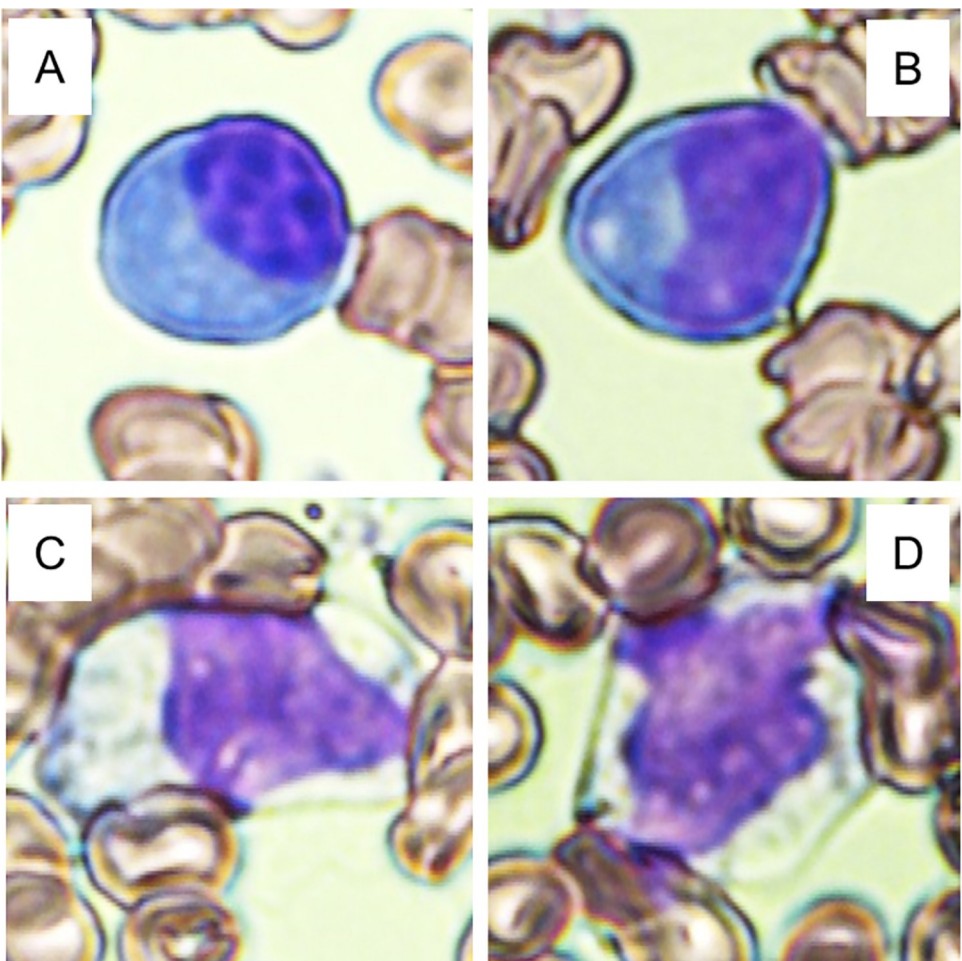

**Fig 1. Representative atypical lymphocytes in the peripheral blood of COVID-19 patients.** All images are of May-Giemsa-stained peripheral blood smears. (A, B) Atypical lymphocytes with condensed chromatin, deep basophilic cytoplasm and eccentric nuclei. (C, D) Atypical lymphocytes with abundant pale cytoplasm and indented nuclei, resembling Downey II cells.

## Laboratory features

A total of 308 blood samples were collected during the study period. The mean interval between each blood test was 3.24 days (2.25–4.00). We chose the peak or nadir values as the representative values to investigate the relationship between the presence of atypical lymphocytes and laboratory test results. Table 2 shows the differences in each test between the patients with and without atypical lymphocytes. The peak values of neutrophils, lymphocytes, LDH and CRP were significantly higher in patients with atypical lymphocytes.

## Disease course in patients with supplemental oxygen

Among the 51 patients who had atypical lymphocytes, 13 received supplemental oxygen therapy. We investigated the details of their clinical courses. Four patients showed a halt then an improvement of respiratory failure after the detection of atypical lymphocytes. In four patients, atypical lymphocytes appeared at the time that oxygen therapy was withdrawn. The other five patients experienced worsening of their condition after the presence of atypical lymphocytes.

**Table 1. Patient characteristics.**

| | Atypical lymphocytes (+) | Atypical lymphocytes (-) | p-value [a] |
|---|---|---|---|
| No. of patients | 51 | 47 | |
| Age (year) | 49.0 (33.5–60.0) | 37.0 (28.0–53.0) | 0.0952 |
| Sex (male) | 35 (68.6%) | 32 (68.1%) | 1.0000 |
| Onset to admission (day) | 5.0 (3.0–8.0) | 4.0 (2.0–5.5) | **0.0214** |
| Symptoms | | | |
| • Fever | 45 (88.2%) | 29 (61.7%) | **0.0042** |
| • Upper respiratory | 11 (21.6%) | 17 (36.2%) | 0.1231 |
| • Lower respiratory | 26 (51.0%) | 16 (34.0%) | 0.1053 |
| • Constitutional | 23 (45.1%) | 23 (48.9%) | 0.8397 |
| • Loose stool | 5 (9.8%) | 7 (14.9%) | 0.5433 |
| • Olfactory/Taste | 10 (19.6%) | 9 (19.1%) | 1.0000 |
| Asymptomatic | 0 (0.0%) | 5 (10.6%) | **0.0226** |
| Comorbidities | | | |
| • Obesity | 4 (7.8%) | 2 (4.3%) | 0.6792 |
| • Diabetes mellitus | 9 (17.6%) | 3 (6.4%) | 0.1249 |
| • Hypertension | 14 (27.5%) | 6 (12.8%) | 0.0838 |
| • Malignancy | 0 (0.0%) | 1 (2.1%) | 0.4796 |
| • Others | 3 (5.9%) [b] | 4 (8.5%) [c] | 0.7071 |
| • No. of comorbidities | 0 (0–1) | 0 (0–0.25) | 0.1453 |
| (maximum) | 3 | 3 | |
| Radiographic evidence of pneumonia | 41 (80.4%) | 20 (42.6%) | < **0.0001** |
| Supplemental oxygen | 13 (25.5%) | 2 (4.3%) | **0.0042** |
| Therapeutic medications | | | |
| • Favipiravir | 8 (15.7%) | 3 (6.3%) | **0.0075** |
| • Corticosteroids [d] | 5 (9.8%) | 0 (0.0%) | |
| • Both | 5 (9.8%) | 1 (2.1%) | |
| • None | 33 (64.7%) | 43 (91.5%) | |
| Clinical outcome | | | |
| • ICU administration | 2 (3.9%) | 1 (2.1%) | 1.0000 |

Data are presented as n (percentage) or the median (interquartile range), unless otherwise stated.

[a] Bolded numbers indicate p < 0.05 according to the Wilcoxon rank-sum test (continuous variables) or Fisher's exact test (ratio of categorical variables).

[b] Including one with dilated cardiomyopathy, one with atrial fibrillation and one with tachycardia-bradycardia syndrome with pacemaker implantation.

[c] Including one with cerebrovascular disease, two with chronic kidney disease and one with chronic hepatitis.

[d] Corticosteroids included methylprednisolone (one patient) and dexamethasone (all others).

However, three of the five patients improved after mild worsening without requirements of intensive care. Two patients suffered severe worsening and needed to enter the ICU. The clinical charts of representative patients are summarized in Fig 5.

We compared the detection date of atypical lymphocytes, peak fraction of atypical lymphocytes and peak number of atypical lymphocytes among these clinical courses. As shown in Fig 6, there were no overt differences in these indices among clinical courses.

## Discussion

This study investigated the dynamics and clinical significance of atypical lymphocytes in the peripheral blood of COVID-19 patients. The results showed that atypical lymphocytes were found in approximately half of the COVID-19 patients; however, it took approximately one

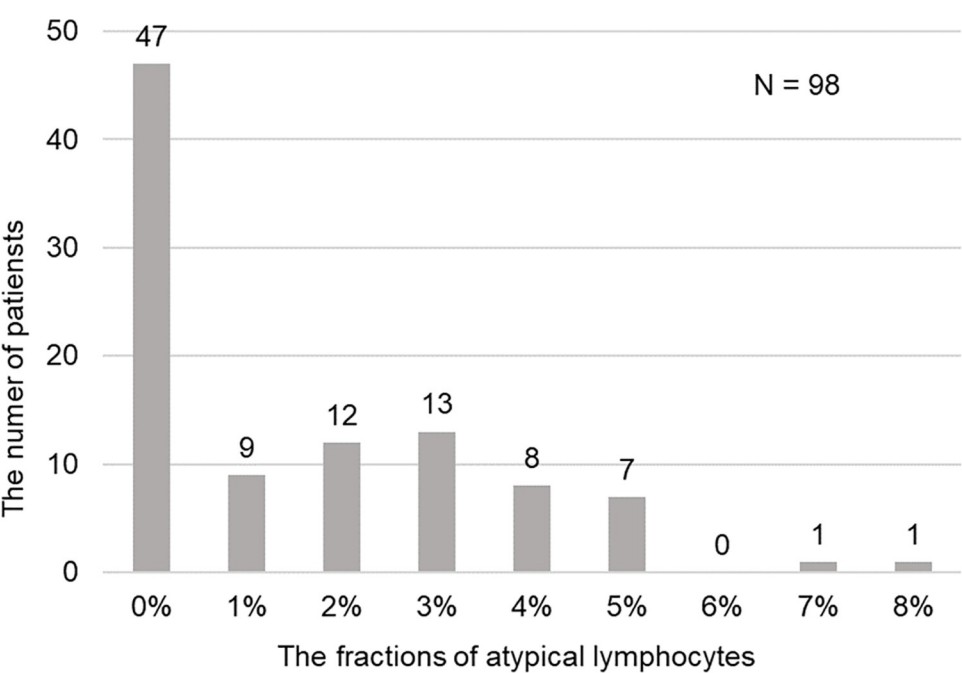

**Fig 2. Distribution of the fractions of atypical lymphocytes.** The number of patients in each fraction of atypical lymphocytes is shown. Each patient was classified according to their peak value. Numbers above bars indicate the actual counts of each fraction.

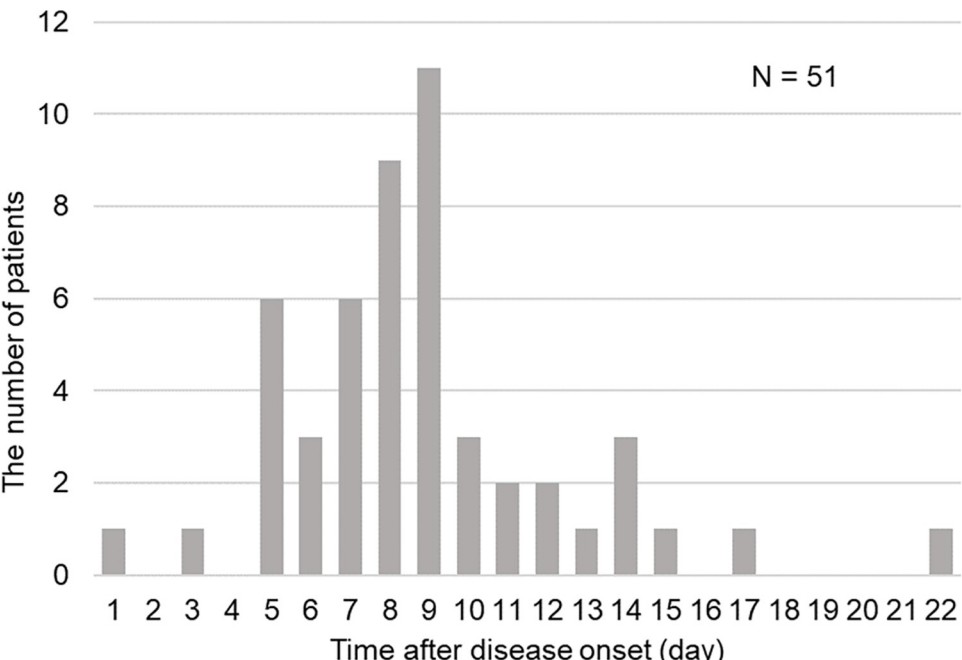

**Fig 3. First appearance of atypical lymphocytes after disease onset.** Patients were counted according to the day of the first identification of atypical lymphocytes after disease onset. The total number of patients who had atypical lymphocytes in their peripheral blood was 51. The median value was eight days.

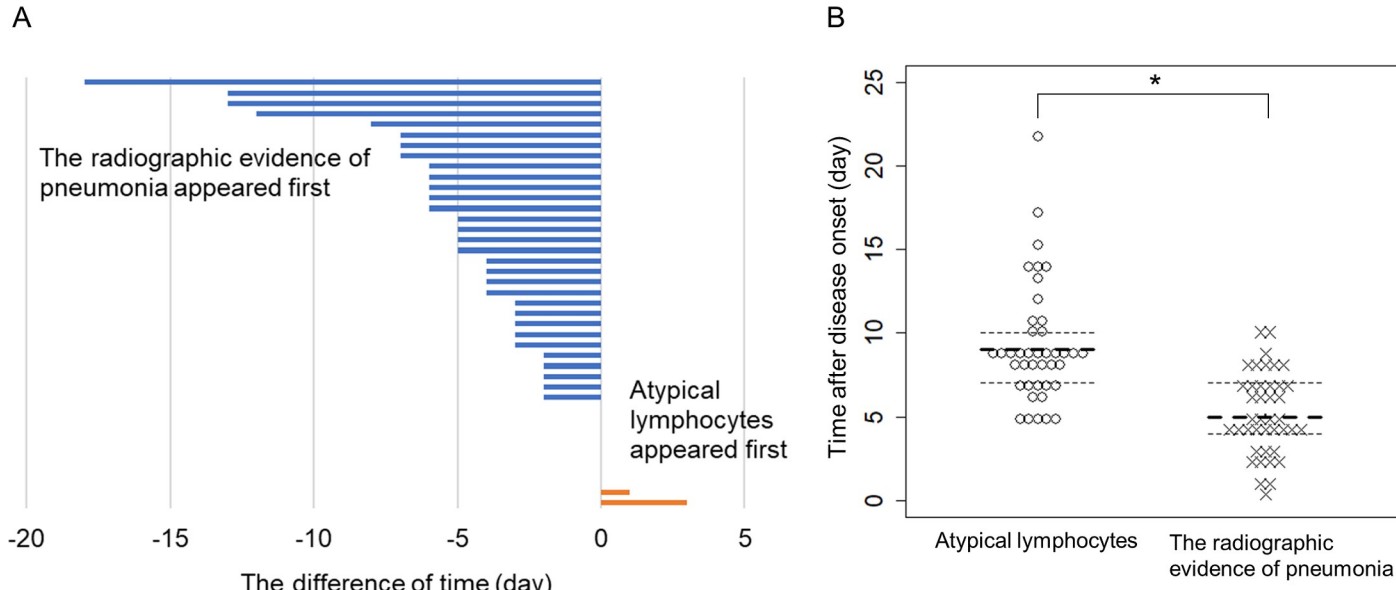

**Fig 4. Comparison of dates on which pneumonia and atypical lymphocytes appeared.** (A) Difference between the days on which radiographic evidence of pneumonia and atypical lymphocytes were detected in each patient. Total number of patients was 41. Bars represent the difference for each patient in days and are arranged in ascending order. A negative value (blue bar) indicates that the radiographic evidence of pneumonia appeared first, and a positive value (orange bar) indicates that atypical lymphocytes appeared first. (B) Beeswarm plot of the first day of detection. The bold dotted line indicates the median, and the thin dotted lines indicate the upper and lower quantiles. * p < 0.0001 by Wilcoxon rank-sum test.

week from disease onset for them to appear. Considering their late appearance, atypical lymphocytes are not useful for the early diagnosis of COVID-19. They are likely to appear after the diagnosis has been made based on another test, such as PCR. However, the presence of atypical lymphocytes is thought to be a distinctive feature of COVID-19 compared to other viral respiratory infections. For example, a study that investigated viral influenza-like illnesses reported that 1/4 of the patients with human metapneumovirus infection, 2/10 of the patients with

**Table 2. Comparison of laboratory values.**

| | Atypical lymphocytes (+) | Atypical lymphocytes (-) | p-value [a] |
|---|---|---|---|
| Peak atypical lymphocytes ($10^6$/L) | 134.0 (93.5–222.0) | 0 | N/A |
| Peak neutrophils ($10^9$/L) | 4.24 (2.74–5.52) | 3.00 (2.49–4.26) | **0.0069** |
| Peak lymphocytes ($10^9$/L) | 2.00 (1.61–2.41) | 1.56 (1.22–1.93) | **0.0004** |
| Nadir lymphocytes ($10^9$/L) | 1.15 (0.79–1.63) | 1.10 (0.86–1.46) | 0.6931 |
| Peak N/L ratio | 2.88 (1.80–5.88) | 2.52 (1.55–4.15) | 0.2642 |
| Nadir Hb (g/L) | 142.0 (131.5–152.0) | 146.0 (139.5–153.5) | 0.2320 |
| Nadir PLT ($10^9$/L) | 186.0 (133.5–210.0) | 193.0 (165.0–221.5) | 0.2765 |
| Peak LDH (U/L) | 249 (195–274.1) | 177 (159–204) | **< 0.0001** |
| Peak T-Bil (μmol/L) | 11.97 (10.26–15.39) | 11.97 (10.26–15.39) | 0.7763 |
| Peak UN (mmol/L) | 5.14 (4.25–6.18) | 4.32 (3.87–5.68) | 0.1026 |
| Peak Creatinine (μmol/L) | 81.33 (66.30–87.52) | 73.37 (62.32–80.44) | 0.1777 |
| Peak CRP (mg/L) | 27.00 (7.55–69.10) | 3.00 (0.55–16.75) | **< 0.0001** |

Abbreviations: N/A, not applicable; N/L ratio, neutrophil to lymphocyte ratio; RBC, red blood cell; Hb, hemoglobin; PLT, platelet. Data are presented as the median (interquartile range) unless otherwise stated.

[a] Bolded number indicates p < 0.05 in the Wilcoxon rank-sum test.

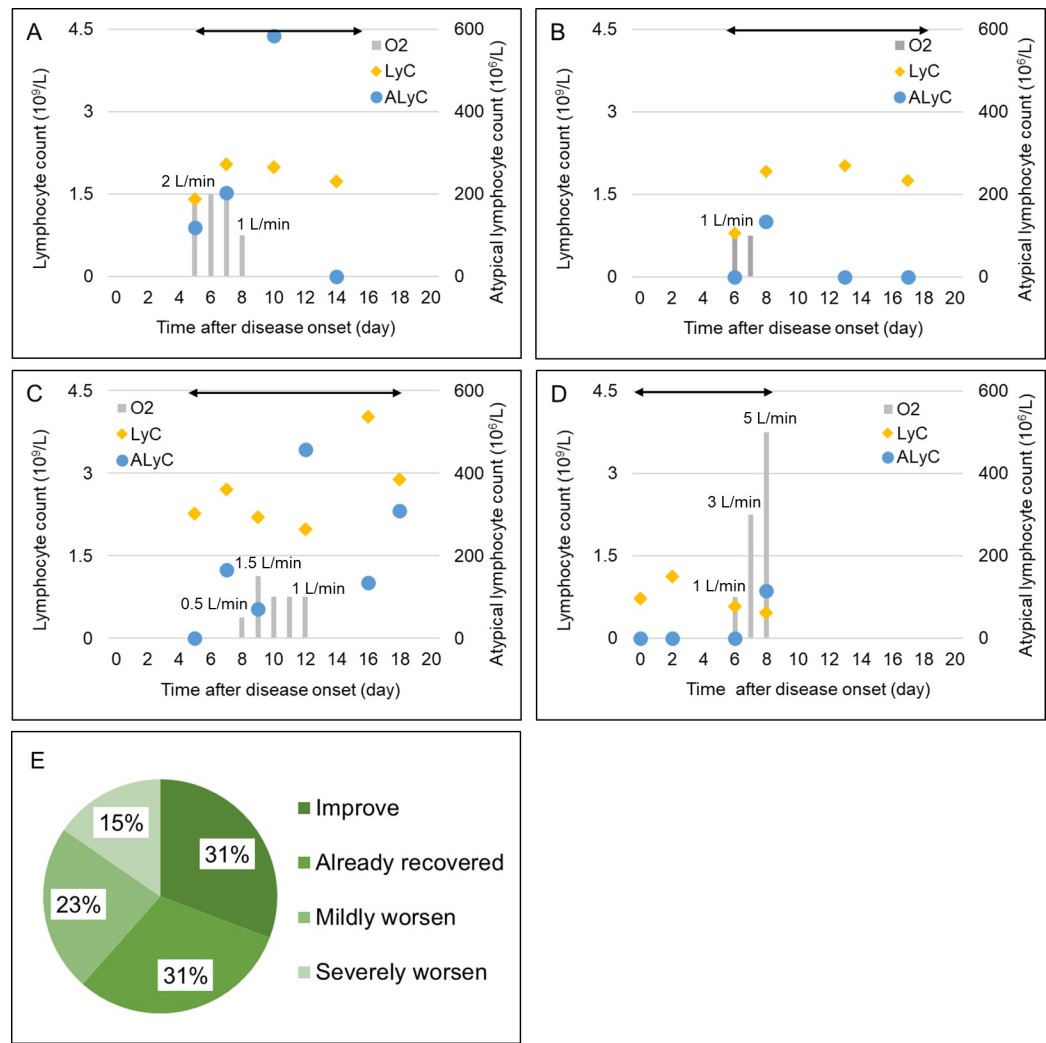

**Fig 5. Clinical courses of representative patients who received supplemental oxygen.** Abbreviations: LyC, lymphocyte count; ALyC, atypical lymphocyte count. (A-D) Time course of lymphocyte count (yellow diamond), atypical lymphocyte count (blue circle) and amount of supplemental oxygen (gray bar) in representative patients. The number above the gray bar demonstrates the amount of oxygen in liters per minute, and double-headed arrows indicate the period of data collection for each patient. (A) A patient who improved after atypical lymphocytes were detected. (B) A patient who had already stopped oxygen therapy when atypical lymphocytes appeared. (C) A patient whose condition worsened after atypical lymphocytes appeared. (D) A patient who worsened when the presence of atypical lymphocytes was detected. This patient was admitted to the ICU on day 8. (E) Proportion of patients in each clinical category. A–D correspond to the categories mentioned above.

rhinovirus/enterovirus infection, 1/16 of the patients with respiratory syncytial virus infection and 3/5 of the patients with human parainfluenza virus-3 infection had atypical lymphocytes; however, the percentages of atypical lymphocytes in these patients ranged from one to three percent [9]. Two studies on SARS show conflicting results: one reported that 15.2% of the patients had atypical lymphocytes [10], and the other reported no patients [11]. Compared to these viral infections, COVID-19 tends to cause more frequent and intensive atypical lymphocytosis. This has implications for our understanding of the nature of COVID-19.

In the context of COVID-19, atypical lymphocytes are regarded as immunologically activated T cells. This idea is supported by the findings of several studies. One study revealed that

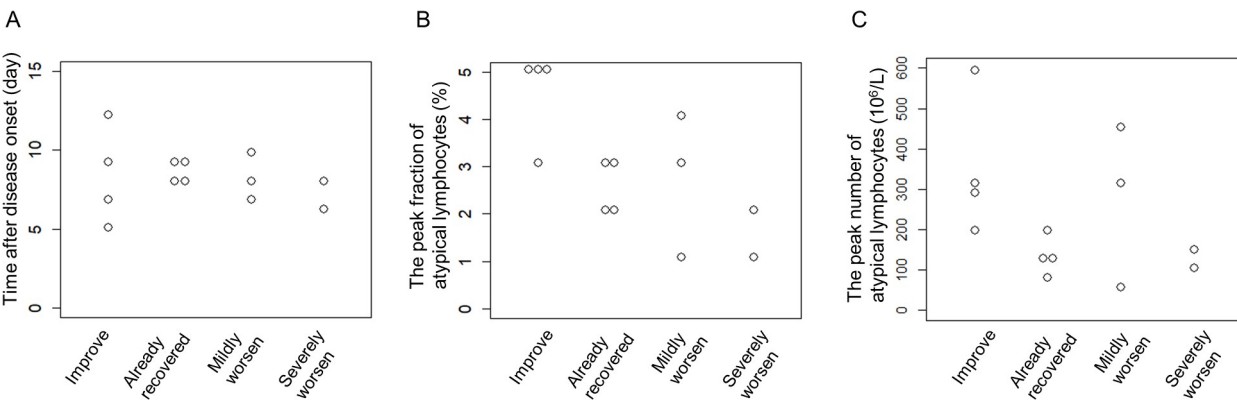

**Fig 6. Comparison of clinical indices among clinical courses.** (A) Comparison of the date on which atypical lymphocytes appeared. (B) Comparison of the peak fraction of atypical lymphocytes. (C) Comparison of the peak number of atypical lymphocytes. Each plot indicates a patient in a group. No significant differences were detected among groups by one-way ANOVA.

patients with large lymphocytes (regarded as atypical lymphocytes) had abundant T cells, and patients with large lymphocytes had significantly more effector memory CD4-positve T cells and CD8-positive T cells than the patient without atypical lymphocytes [12]. Another report that investigated BAL samples from COVID-19 patients demonstrated that atypical lymphocytes in BAL samples were virtually all CD3-positive T cells, and a significant proportion of T cells (40–80%) in BAL samples expressed activation markers such as CD38, HLA-DR and CD25 [6]. Based on these results, we assume that the presence of atypical lymphocytes reflects the activation of T cells in response to infection with SARS-CoV-2.

In regard to T cell dynamics, it was reported that the levels of T cells were depressed seven to nine days after onset in patients with severe cases, while T cell counts were maintained in those with mild cases [13]. It was also reported that the initially reduced number of T cells in patients with severe COVID-19 rose approximately 15 days after onset [14]. These results suggest that the prolonged period from disease onset to the presence of atypical lymphocytes reflected the characteristics of lymphocyte dynamics in patients with COVID-19.

On the other hand, the appearance of atypical lymphocytes seemed to be associated with a favorable disease course, such as a decrease in or cessation of the need for supplemental oxygen, as was observed in approximately two-thirds of the patients in this study. This result was consistent with a previous study that suggested an association between atypical lymphocytes and mild disease [8]. A previous observational study showed that CD8-positive T cell counts were reduced in patients with severe COVID-19 and that they recovered in patients who responded to treatment [15], also suggesting the association of T cells with disease severity. It was shown that atypical lymphocyte count of BAL samples inversely correlated with the length of hospital stay and length on mechanical ventilation in COVID-19 patients [6]. These results seem to indicate that the activation of T cells plays an important role in recovery from COVID-19. Our research seems to reinforce this idea for two reasons: (1) pulmonary infiltration tended to precede the observation of atypical lymphocytes in many patients, and (2) although it correlated with an increasing incidence of radiographic evidence of pneumonia, the presence of atypical lymphocytes did not necessarily lead to clinical deterioration. However, we were not able to demonstrate the proportional relationship between the fraction/absolute number of atypical lymphocytes and disease severity. This may be thought to prove that the activation of T cells was not related to recovery from COVID-19. Our explanation is that disease severity is not merely a cause or a result of immunological activation. Though severe

infection may activate T cells and increase atypical lymphocytes, less activated immunity may lead to worsening infection without activated T cells. When we considered the interaction of virus and immunity, it was not surprising that the number of atypical lymphocytes did not have a clear relationship with disease severity. At any rate, studies with much larger cohorts, including patients with more severe disease necessitating intubation, are needed to reveal the details of the pathophysiology and clinical significance. This should be explored in future research.

This study has several limitations. This is a single-center retrospective study and therefore subject to selection bias and information bias. The study design and public health policy in Japan necessitated the exclusion of patients with severe COVID-19 from the study, which may have influenced the low prevalence of comorbidities, given that COVID-19 tends to be exacerbated in patients with comorbidities. In this observational study, the presence of atypical lymphocytes or radiographic evidence of pneumonia could have been missed because we sometimes did not repeatedly examine patients with mild cases who did not have abnormalities on the first examination. To overcome these biases, further investigations at several institutions with ICUs need to be conducted.

## Conclusion

In this study, we demonstrated that atypical lymphocytes frequently appeared in patients with COVID-19. Patients with atypical lymphocytes were more likely to have radiographic evidence of pneumonia and need supplemental oxygen; however, two-thirds of them had an improved clinical course after the presence of atypical lymphocytes.

## Supporting information

**S1 Appendix. Clinical data of patients.**
(XLSX)

## Acknowledgments

We acknowledge all the healthcare workers in Kashiwa Municipal Hospital who were involved in the clinical treatment of COVID-19 and all patients in our study.

## Author Contributions

**Conceptualization:** Jun Sugihara, Masafumi Doi.

**Investigation:** Jun Sugihara, Sho Shibata, Masafumi Doi, Takuya Shimmura, Shinichiro Inoue, Osamu Matsumoto, Hiroyuki Suzuki, Ayaka Makino.

**Project administration:** Jun Sugihara.

**Visualization:** Osamu Matsumoto.

**Writing – original draft:** Jun Sugihara.

**Writing – review & editing:** Sho Shibata, Yasunari Miyazaki.

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
