## [Decision Letter · Decision Letter 0]

3 Aug 2021

PONE-D-21-21070

Atypical lymphocytes in the peripheral blood of COVID-19 patients: a prognostic factor for the clinical course of COVID-19.

PLOS ONE

Dear Dr. Sugihara,

Thank you for submitting your manuscript to PLOS ONE. After careful consideration, we feel that it has merit but does not fully meet PLOS ONE’s publication criteria as it currently stands. Therefore, we invite you to submit a revised version of the manuscript that addresses the points raised during the review process.

The reviewers agreed that this is an interesting study, however, there are a few comments that need to be addressed.  Please see reviewers' insightful comments below.  Personally, on a more detail level, I also have a few comments that need to be addressed. 1. Line 102 - 104:  Are these daily occurrence? If not daily, how often does this test performed.  2. Expand the explanation of Fig. 2.  For example:  47 out of 98 patients (48%) blood smear did not show atypical lymphocytes.  There are 9 patients with 1% atypical lymphocytes.  12 and 13 patients had 2% and 3% atypical lymphocytes respectively, while 8 and 7 patients had 4 and 5% atypical lymphocytes.  Higher fraction (7 & 8%) of atypical lymphocytes was observed in only 2 patients (one for each fraction).  This bags the question whether the fraction of atypical lymphocytes associated with disease severity.  The n of this study is probably too small to have an accurate estimate, but discuss may be warranted.  3. Expand explanation is also needed for Fig. 3.  For example.  The majority of the patients with atypical lymphocytes was detected 5 - 9 days after disease onset and peaked at 9 days.

In addition, the quality of the language needs to be improved.  Please have a fluent, preferably native, English-language speaker thoroughly copyedit your manuscript for language usage, spelling, and grammar.  

We look forward to receiving your revised manuscript.

Kind regards,

Baochuan Lin, Ph.D.

Academic Editor

PLOS ONE

Journal Requirements:

“This research was supported by AMED under Grant Number 21jk0210034h0002 to S.S.”

We note that you have provided funding information within the Acknowledgements. Please note that funding information should not appear in the Acknowledgments section or other areas of your manuscript. We will only publish funding information present in the Funding Statement section of the online submission form.

“This study is funded by Japan Agency for Medical Research and Development.

S. S. received this award.

Grant number is 21jk0210034h0002.

URL of the funder is https://www.amed.go.jp/en/index.html

The funder had no role in study design, data collection and analysis, decision to publish, or preparation of the manuscript.”

Reviewers' comments:

Reviewer's Responses to Questions

**Comments to the Author**

1. Is the manuscript technically sound, and do the data support the conclusions?

Reviewer #1: Yes

Reviewer #2: Yes

2. Has the statistical analysis been performed appropriately and rigorously? 

Reviewer #1: Yes

Reviewer #2: Yes

3. Have the authors made all data underlying the findings in their manuscript fully available?

Reviewer #1: Yes

Reviewer #2: Yes

4. Is the manuscript presented in an intelligible fashion and written in standard English?

Reviewer #1: Yes

Reviewer #2: Yes

5. Review Comments to the Author

Reviewer #1: The current manuscript focused on the investigation of the dynamics of atypical lymphocytes in COVID-19 patients to estimate clinical significance in Japanese population. The conclusion is consistent with previous studies, which provide additional evidence of the values of atypical lymphocytes in COVID-19 in different populations. The manuscript is well written. I have a few minor suggestions as below:

1. The pictures in Figure 1 could be improved. The atypical lymphocytes can be enlarged with few surrounding red cells.

2. A recent study by Gelarden et al (Human Pathology, 2021 Jul; 113:92-103) reported the significance of atypical lymphocytes in bronchoalveolar lavage from COVID-19 patients that can be correlated with clinical outcomes. It would be better that the authors considering discussing their findings with this paper.

Reviewer #2: This is a good study providing information on timely requirements.

In the study, you have showed the atypical lymphocytes as a prognostic factor. Can it be used as a prognostic marker or a predictive marker for prognosis? if so on which day of identifying atypical lymphocyte could suggest pneumonia and related complications among covid-19 patients? This could be then further analysed to provide sensitivity and specificity of using atypical lymphocytes as a prognostic marker. Also address the following:

01. Number/percentage of atypical lymphocyte that could act as a prognostic factor on which day?

02. Gap between detection of atypical lymphocyte and pneumonia

03. statistical relation between oxygen therapy and predictive role of atypical lymphocyte in it.

6. PLOS authors have the option to publish the peer review history of their article (what does this mean?). If published, this will include your full peer review and any attached files.

Reviewer #1: No

Reviewer #2: **Yes: **Roshan Niloofa

---

## [Author Response · Author response to Decision Letter 0]

14 Sep 2021

Dear Dr. Baochuan Lin,

 Thank you very much for your e-mail and review of the manuscript (PONE-D-21-21070) that we sent on June 28, 2021. We thank you and the two reviewers for providing constructive comments regarding the improvement of the original manuscript.

Here, we are sending a PDF file of our revised manuscript. All changes have been made in response to your suggestions, and itemized responses to the individual reviewer’s comments are also attached.

Response to academic editor:

Q1. Line 102 - 104: Are these daily occurrences? If not daily, how often does this test performed.

Response: These tests were not performed daily, and we calculated the frequency and revised it in Line 196–197.

Q2. Expand the explanation of Fig. 2. For example: 47 out of 98 patients (48%) blood smear did not show atypical lymphocytes. There are 9 patients with 1% atypical lymphocytes. 12 and 13 patients had 2% and 3% atypical lymphocytes respectively, while 8 and 7 patients had 4 and 5% atypical lymphocytes. Higher fraction (7 & 8%) of atypical lymphocytes was observed in only 2 patients (one for each fraction).

This bags the question whether the fraction of atypical lymphocytes associated with disease severity. The n of this study is probably too small to have an accurate estimate, but discuss may be warranted.

Response: We thank the editor for a valuable suggestion. The explanation was expanded as indicated. We also analyzed the relationship between the fraction and disease severity, and it seems that the fraction/number of atypical lymphocytes is not significantly associated with disease severity. In brief, we assumed that there would be two directions of interaction: severe infection provokes an immune reaction and leads to an increase in atypical lymphocytes; otherwise, an inappropriate immune response results in severe disease and a low number of atypical lymphocytes. We appended a detailed discussion in Lines 281–287.

Q3. Expand explanation is also needed for Fig. 3. For example. The majority of the patients with atypical lymphocytes was detected 5 - 9 days after disease onset and peaked at 9 days

Response: We thank the editor for providing this helpful advice. The explanation is expanded as indicated.

Q4. The quality of the language needs to be improved. Please have a fluent, preferably native, English-language speaker thoroughly copyedit your manuscript for language usage, spelling, and grammar.

Response: The paper has been edited and rewritten by an experienced scientific editor, who has improved the grammar and stylistic expression of the paper.

Response to journal Requirements:

Q1. Please ensure that your manuscript meets PLOS ONE's style requirements, including those for file naming.

Response: We checked our manuscript and attached files thoroughly.

Q2. Please remove any funding-related text from the manuscript and let us know how you would like to update your Funding Statement. Currently, your Funding Statement reads as follows:

“This study is funded by Japan Agency for Medical Research and Development.

S. S. received this award.

Grant number is 21jk0210034h0002.

URL of the funder is https://www.amed.go.jp/en/index.html

The funder had no role in study design, data collection and analysis, decision to publish, or preparation of the manuscript.”

Response: As indicated, funding-related text was removed from the Acknowledgments. Funding Statement did not need to be amended.

Q3. Upon re-submitting your revised manuscript, please upload your study’s minimal underlying data set as either Supporting Information files or to a stable, public repository and include the relevant URLs, DOIs, or accession numbers within your revised cover letter. For a list of acceptable repositories, please see http://journals.plos.org/plosone/s/data-availability#loc-recommended-repositories. Any potentially identifying patient information must be fully anonymized.

Response: We prepared our study's minimal underlying dataset and uploaded it as a Supporting Information file named "S1 Appendix. Clinical data of patients".

Q4. Please review your reference list to ensure that it is complete and correct. If you have cited papers that have been retracted, please include the rationale for doing so in the manuscript text, or remove these references and replace them with relevant current references. Any changes to the reference list should be mentioned in the rebuttal letter that accompanies your revised manuscript. If you need to cite a retracted article, indicate the article’s retracted status in the References list and also include a citation and full reference for the retraction notice.

Response: We checked the reference list of our manuscript, and no cited papers were retracted.

Response to reviewer #1:

Q1. The pictures in Figure 1 could be improved. The atypical lymphocytes can be enlarged with few surrounding red cells.

Response: We thank the reviewer for pointing out how to improve this figure. Figure 1 was revised as indicated.

Q2. A recent study by Gelarden et al (Human Pathology, 2021 Jul; 113:92-103) reported the significance of atypical lymphocytes in bronchoalveolar lavage from COVID-19 patients that can be correlated with clinical outcomes. It would be better that the authors considering discussing their findings with this paper.

Response: We thank you for introducing us to an important paper, which should be mentioned in our manuscript. We read it closely and have changed the text throughout the Discussion in response to its valid points.

Response to reviewer #2:

Q1. In the study, you have showed the atypical lymphocytes as a prognostic factor. Can it be used as a prognostic marker or a predictive marker for prognosis? if so on which day of identifying atypical lymphocyte could suggest pneumonia and related complications among covid-19 patients? This could be then further analysed to provide sensitivity and specificity of using atypical lymphocytes as a prognostic marker.

Response: Although we analyzed atypical lymphocyte dynamics and pneumonia progression, it was hard to clarify the relationship because of the small number of patients who underwent CT scans several times during the follow-up period (40 out of 51 patients did not undergo CT scans after atypical lymphocytes presented in their peripheral blood). Regarding related complications, respiratory failure is the one we could argue on the basis of our data. As calculated by the total number of patients shown in Table 1, the sensitivity and specificity of atypical lymphocytes for oxygen therapy were 0.87 and 0.54, respectively. However, when we further analyzed several clinical indices, as shown in Figure 6, we were not able to show a clear relationship. Considering the results of our investigation, we concluded that atypical lymphocytes correlated with oxygen therapy and relatively favorable prognosis. This idea seems consistent with the presumption that the presence of atypical lymphocytes is a response to viral infection.

Q2. Number/percentage of atypical lymphocyte that could act as a prognostic factor on which day?

Response: We analyzed our data but was not able to detect a significant relationship between the number/percentage of atypical lymphocytes and prognosis. We assumed that this was because there were two meanings of atypical lymphocytes; they indicate both active infection and a lively response of the immune system. We appended a detailed discussion in Lines 281–287. We think the existence of atypical lymphocytes is rather important as a prognostic factor.

Q3. Gap between detection of atypical lymphocyte and pneumonia

Response: If you mean the gap in the dates, we showed these results in Figure 4A. and Lines 178–182.

Q4. statistical relation between oxygen therapy and predictive role of atypical lymphocyte in it.

Response: Please see the response to question #1 above.

Figures and Table:

To improve the manuscript according to the reviewers’ comments, we have changed the figure composition as described below:

Fig. 1 has been revised.

Fig. 5 has been revised.

A new figure has been inserted as Fig 6.

We believe that we have addressed the comments from the academic editor and both reviewers. We look forward to hearing from you regarding our submission. We would be glad to respond to any further questions and comments that you may have. Thank you for your generous consideration.

Sincerely yours,

Jun Sugihara, MD

---

## [Decision Letter · Decision Letter 1]

20 Oct 2021

PONE-D-21-21070R1Atypical lymphocytes in the peripheral blood of COVID-19 patients: a prognostic factor for the clinical course of COVID-19.PLOS ONE

Dear Dr. Sugihara,

Thank you for submitting your manuscript to PLOS ONE. After careful consideration, we feel that it has merit but does not fully meet PLOS ONE’s publication criteria as it currently stands. Therefore, we invite you to submit a revised version of the manuscript that addresses the points raised during the review process.

The reviewers agreed that the revised manuscript is scientifically sound, however, there are a few minor issues that still need to be addressed.  Please see specific comments below. Specific comments:1. Line 147 - 148:  The sentence "There were 9 patients with 1% atypical lymphocytes" was repeated, please delete one.2. Line 282 - 283:  "This may be thought to prove otherwise." Not sure what the authors wish to convey, please rephrase for clarification.3. Line 284:  Should "Thought" be "Though"?4. Line 286:  Suggest changing "seemed" to "was".5. Line 296: Suggest changing "occasionally" to "sometimes"6. Line 304:  Delete "to"

We look forward to receiving your revised manuscript.

Kind regards,

Baochuan Lin, Ph.D.

Academic Editor

PLOS ONE

Journal Requirements:

Reviewers' comments:

Reviewer's Responses to Questions

**Comments to the Author**

1. If the authors have adequately addressed your comments raised in a previous round of review and you feel that this manuscript is now acceptable for publication, you may indicate that here to bypass the “Comments to the Author” section, enter your conflict of interest statement in the “Confidential to Editor” section, and submit your "Accept" recommendation.

Reviewer #1: All comments have been addressed

Reviewer #2: All comments have been addressed

2. Is the manuscript technically sound, and do the data support the conclusions?

Reviewer #1: Yes

Reviewer #2: Yes

3. Has the statistical analysis been performed appropriately and rigorously? 

Reviewer #1: Yes

Reviewer #2: Yes

4. Have the authors made all data underlying the findings in their manuscript fully available?

Reviewer #1: Yes

Reviewer #2: Yes

5. Is the manuscript presented in an intelligible fashion and written in standard English?

Reviewer #1: Yes

Reviewer #2: Yes

6. Review Comments to the Author

Reviewer #1: The authors adequately addressed my comments. I have no more concerns. The study would be impactful for the clinical practice on COVID19.

Reviewer #2: Given comments have been addressed by the authors and the manuscript is generally improved. Well written and a timely paper.

7. PLOS authors have the option to publish the peer review history of their article (what does this mean?). If published, this will include your full peer review and any attached files.

Reviewer #1: No

Reviewer #2: No

---

## [Author Response · Author response to Decision Letter 1]

24 Oct 2021

Response to comments:

1. Line 147 - 148: The sentence "There were 9 patients with 1% atypical lymphocytes" was repeated, please delete one.

Response: We thank the editor for kind advice. The duplicative sentence was deleted as indicated.

2. Line 282 - 283: "This may be thought to prove otherwise." Not sure what the authors wish to convey, please rephrase for clarification.

Response: We thank the editor for pointing out an insufficient sentence in our manuscript. We rephrased this sentence as "This may be thought to prove that the activation of T cells was not related to recovery from COVID-19".

3. Line 284: Should "Thought" be "Though"?

Response: Yes. We thank the editor for kind advice, and correct the word as indicated. 

4. Line 286: Suggest changing "seemed" to "was".

Response: We thank the editor for this helpful suggestion, and correct the sentence as indicated. 

5. Line 296: Suggest changing "occasionally" to "sometimes"

Response: We thank the editor for this helpful suggestion, and correct the sentence as indicated.

6. Line 304: Delete "to"

Response: We thank the editor for kind advice. The sentence was corrected as indicated.

---

## [Editor Report · Decision Letter 2]

29 Oct 2021

Atypical lymphocytes in the peripheral blood of COVID-19 patients: a prognostic factor for the clinical course of COVID-19.

PONE-D-21-21070R2

Dear Dr. Sugihara,

We’re pleased to inform you that your manuscript has been judged scientifically suitable for publication and will be formally accepted for publication once it meets all outstanding technical requirements.

Kind regards,

Baochuan Lin, Ph.D.

Academic Editor

PLOS ONE
---

## [Editor Report · Acceptance letter]

4 Nov 2021

PONE-D-21-21070R2 

Atypical lymphocytes in the peripheral blood of COVID-19 patients: a prognostic factor for the clinical course of COVID-19. 

Dear Dr. Sugihara:

I'm pleased to inform you that your manuscript has been deemed suitable for publication in PLOS ONE. Congratulations! Your manuscript is now with our production department. 

Kind regards, 

on behalf of

Dr. Baochuan Lin 

Academic Editor

PLOS ONE